# Genome-Wide Identification, Phylogenetic and Expression Analysis of the B-Box Gene Family in the Woodland Strawberry (*Fragaria vesca*)

**Dong Xu** [1], **Hongkun Wang** [1], **Xiaotian Feng** [1], **Yuqing Ma** [1], **Yirui Huang** [1], **Yushan Wang** [1], **Jing Ding** [1], **Hong Chen** [2,*] and **Han Wu** [1,*]

1   State Key Laboratory of Crop Genetics & Germplasm Enhancement and Utilization, College of Horticulture, Nanjing Agricultural University, Nanjing 210095, China; 2021104068@stu.njau.edu.cn (D.X.); 2022804269@stu.njau.edu.cn (H.W.); 14119304@njau.edu.cn (X.F.); 9201410801@stu.njau.edu.cn (Y.M.); 9201410825@stu.njau.edu.cn (Y.H.); 9201410803@stu.njau.edu.cn (Y.W.); jding@njau.edu.cn (J.D.)
2   Jiangsu Key Laboratory for the Research and Utilization of Plant Resources, Institute of Botany, Jiangsu Province and Chinese Academy of Sciences, Nanjing 210014, China
*   Correspondence: wuhan@njau.edu.cn (H.W.); chenhong@cnbg.net (H.C.)

**Abstract:** The strawberry (*Fragaria* × *ananassa* Duch.) is an important horticultural crop that is widely grown all over the world. Its sweetness, aroma, nutritional value and bright color make it popular. The woodland strawberry (*Fragaria vesca*) is a model plant for studying non-climacteric fruits because its respiration rate does not change significantly during fruit ripening. The B-box (BBX) protein family is made up of zinc-finger transcription factors important in plant growth and development. In this study, we identified 22 *FveBBX* genes from the newly released woodland strawberry genome database by comprehensive bioinformatics analysis. Phylogenetic analysis divided these *FveBBX* genes into five subfamilies. A promoter cis-acting element analysis detected 29 elements related to plant development, light response, abiotic stress and hormone response in the promoter of *FveBBX* genes. According to transcriptome data, relatively few *BBX* genes had tissue-specific expression, with examples including *FveBBX12*, which was expressed only in pre-fertilization cortex and pitch, and *FveBBX19*, which was specifically expressed in mature anthers. During fruit ripening, the expressions of eight *FveBBX* genes decreased by more than two-fold, and three *FveBBX* gene expressions increased more than two-fold both in "Ruegen" and "Yellow Wonder". After cold and heat stresses, around half of the *FveBBX* genes displayed altered expression, especially *FveBBX16* which showed an 8.3-fold increase in expression after heat treatment, while *FveBBX14* showed at least an 11-fold decrease in expression after cold treatment. According to the result of quantitative real-time PCR (qRT-PCR), *FveBBX* genes' expression differed depending on the photoperiod. Notably, *FveBBX7* gene expression was the opposite during the first 16 h of the long-day (LD) and short-day (SD) conditions. This study provides helpful information for further research on *BBX* gene activity of the woodland strawberry in plant growth and development and adaptation to temperature and photoperiod environmental conditions.

**Keywords:** genome-wide identification; expression pattern; heat and cold; photoperiod; B-box proteins; woodland strawberry

## 1. Introduction

The strawberry (*Fragaria* × *ananassa* Duch.) is a popular fruit crop cultivated worldwide that has a complex heterozygous octaploid genome, hampering genetic transformation. As one of the diploid progenitors of the octaploid cultivated strawberry, the woodland strawberry (*Fragaria vesca*) has a relatively simple genome and stable transformation system, which makes it a model plant for studying rosaceae fruit trees and non-climacteric fruits [1].

Transcription factors are an important kind of protein in higher plants that can bind target DNA, interact with other proteins and ensure that the target gene is expressed at a

specific intensity in a specific time and space [2]. The zinc-finger transcription factors have a local peptide structure required for their function, which is formed by cysteine and/or histidine in combination with zinc atoms. Zinc-finger transcription factors are involved in many important biological processes in plants [2]. Among them, B-box (BBX) proteins have received increasingly widespread attention due to their functional diversity [3].

BBX proteins have a representative feature, the B-box structural domain, which is found in proteins of numerous multicellular species and certain unicellular eukaryotes [4]. Two types of B-box domains, B-box1 (B1) and B-box2 (B2), are located near the amino terminus of BBX proteins. The presence of B-box domains is thought to be potentially involved in DNA binding, protein–protein interactions and transcriptional regulation [5]. Certain BBX proteins have a CCT (Constans, co-like, timing of CAB1: TOC1) structural domain at the carboxyl terminus [5]. The CCT structural domain plays an important role in transcriptional regulation and in the localization of BBX proteins in the nucleus [6]. BBX proteins are divided into five subfamilies. Proteins in subfamily I and II contain two B-box structural domains and a CCT structural domain (B1 + B2 + CCT), and those in subfamily I also contain a VP motif. Proteins in subfamily III have one B-box and one CCT structural domain (B1 + CCT). Proteins in subfamily IV contain only two B-box structural domains (B1 + B2), while subfamily V contains BBX proteins with only one B-box structural domain [7]. In *Arabidopsis*, 32 *AtBBX* genes were renamed and identified [3]. In addition, 24, 64, 37, 16, 30, 31 and 30 *BBX* genes were identified in grape [8], apple [9], pear [10], sweet cherry [11], potato [12], tomato [13] and rice [14], respectively, and these *BBX* genes were grouped into five subfamilies similar to *Arabidopsis*. Moreover, all five *BBX* subfamilies evolved independently during plant evolution [7].

*Constans/AtBBX1* (*CO*) in *Arabidopsis* is the first identified gene of the *BBX* family, which plays a significant role in controlling flowering [3]. AtCO is associated with flowering locus T (FT) via the CCT structural domain and activates transcription and initiates the flowering process [15]. In addition, *AtBBX5* [16], *AtBBX6* [17] *AtBBX7* [18] and *AtBBX10* [19] are involved in the regulation of flowering in *Arabidopsis*; AtBBX6 promotes, whereas the others suppress, flowering.

Photomorphogenesis in *Arabidopsis* is also strongly associated with *BBX* genes, for example, AtBBX19 interacts with CO and thus negatively regulates floral transition and flowering [20]. *AtBBX24, AtBBX25* [21], *AtBBX28* [22], *AtBBX29* [23], *AtBBX30* and *AtBBX31* [24] also negatively regulate the flowering process of *Arabidopsis*. Among them, AtBBX28 and AtBBX29 form a feedback pathway with AtBBX30 and AtBBX31 via elongated hypocotyl 5 (HY5). AtBBX28 and AtBBX29 trigger HY5 binding to the promoters of *AtBBX30* and *AtBBX31*, while AtBBX30 and AtBBX31 associate with the promoter regions of *AtBBX28* and *AtBBX29* to promote the expression of these genes [24]. On the other hand, *AtBBX21* [25], *AtBBX22* [26] and *AtBBX23* [27] play an active role in the photomorphogenesis of *Arabidopsis*. Interestingly, AtBBX4 interacts with AtBBX32 to regulate *FT* expression and affect flowering [28]. Certain *BBX* genes are closely related to the shade-avoidance response and branching of plants. For example, overexpression of *Arabidopsis AtBBX16* results in increased branching and an enhanced shade-avoidance response [29].

*BBX* genes' promoters include a number of cis-acting elements that respond to plant hormone signals and combat abiotic stress [30]. For example, ABA treatment induces the expression of *AtBBX5* in *Arabidopsis,* which responds to salt stress through ABA-dependent signaling pathways [31]. In *Arabidopsis*, *AtBBX18* expression is negatively correlated with heat tolerance, and its overexpression reduces the tolerance of transgenic plants to high temperature [32]. AtBBX24 in *Arabidopsis* can bind to the homolog of the MYB transcription factor, and its overexpression enhances salt tolerance [33]. Exogenous expression of *CpBBX19* in the Chinese plum enhances the tolerance of *Arabidopsis* to salt and drought stress [34]. In addition, silencing of *SlBBX7*, *SlBBX9* and *SlBBX20* in tomato enhances the effect of cold stress [13]. Transgenic lines with suppressed *CmBBX24* expression are less tolerant to low temperature and drought stress than the wild type [35].

To date, the *BBX* transcription factor family in the woodland strawberry has been identified genome-wide, but the expression patterns in different tissues and their response to environmental stresses are limited. Therefore, in this study, we identified *FveBBX* gene family members and characterized them at the molecular level. Subsequently, we analyzed the expression of *FveBBX* genes in different tissues and their response to cold and heat treatments. At the same time, qRT-PCR was used to study the expression changes in certain *BBXs* in leaves under different photoperiod conditions. Our results will lay the foundation for further understanding the function of strawberry *BBX* genes.

## 2. Materials and Methods

### 2.1. Identification of BBX Gene Family Members in the Strawberry

The *Arabidopsis BBX* gene family has been well studied, and their protein databases and annotation information were downloaded from the TAIR website (https://www.arabidopsis.org/ Accessed on 16 April 2022). The latest woodland-strawberry-related database was downloaded from GDR (https://www.rosaceae.org/species/fragaria/fragaria_vesca Accessed on 16 April 2022). We used the "blast + hmmer + pfamscan" model to search for *BBX* genes in the woodland strawberry. First, a blast search of the woodland strawberry protein database was performed using 32 *AtBBX* genes of Arabidopsis as representative sequences. Second, the hidden Markov model (HMM) for the conserved B-box domain (pfam00643) was downloaded from the Pfam website (http://pfam.xfam.org/ Accessed on 16 April 2022), and the HMM was used to search the woodland strawberry protein database. Both of these steps were performed in TBtools1.116 software [36]. Finally, the two results were combined and submitted to the Pfamscan website (https://www.ebi.ac.uk/Tools/pfa/pfamscan/ Accessed on 22 May 2022) to identify conserved structural domains with an E-value cut-off of $1 \times 10^{-5}$. The SMART (https://smart.embl-heidelberg.de/ Accessed on 22 May 2022) and CDD (https://www.ncbi.nlm.nih.gov/Structure/cdd/wrpsb.cgi Accessed on 22 May 2022) databases were used to identify the B-box and CCT structural domains of those genes. The molecular weights (MW) and isoelectric points (pI) of FveBBX proteins were calculated from the ExPASy website (https://www.expasy.org/resources/compute-pi-mw Accessed on 23 May 2022).

### 2.2. Gene Structure and Motif Analysis of FveBBX Genes

Woodland strawberry genome annotation files downloaded from the GDR website were submitted to the GSDS website (http://gsds.gao-lab.org/index.php accessed on 23 May 2022) for exon and intron identification. The conservative motif was identified by the MEME website (https://meme-suite.org/meme/ accessed on 23 May 2022) [37], and the maximum number of motifs was set to 20 on the basis that all other parameters are set to default. Furthermore, TBtools1.116 software was used to visualize conserved structural domains and motifs.

### 2.3. Construction of FveBBX Genes Phylogenetic Tree

All phylogenetic trees were constructed based on protein sequences. Multiple sequence comparisons of woodland strawberry BBX protein sequences were performed using ClustalW in BioEdit (Borland, Austin, TX, USA), and phylogenetic trees were constructed in MEGE7.0 (Mega Limited, Auckland, New Zealand) using the neighbor-joining method with bootstrap test repeated 1000 times. ClustalW in MEGA7.0 was used to compare the full length of BBX proteins of strawberry and *Arabidopsis*, the conserved sequences of B-box1, B-box2 and CCT domains of strawberry, and to construct a phylogenetic tree using the neighbor joining method with default parameters. In addition, the alignment of conserved sequences of B-box1, B-box2 and CCT structural domains is generated in Jalview2.10.4b1 (Elixir, London, UK) using the default Muscle.

### 2.4. Cis-Element Prediction for FveBBX Gene Promoters

The promoter sequences of 2000 bp upstream of *FveBBX* genes' transcription start site were extracted by TBtools software. The online tool PlantCARE (http://bioinformatics.psb.ugent.be/webtools/plantcare/html/ accessed on 26 May 2022) was used to analyze and annotate putative cis-elements in the promoter sequences [38].

### 2.5. Transcriptome Data Analysis

The woodland strawberry "Hawaii 4" is grown in the greenhouse of Nanjing Agricultural University. First, the seeds of "Hawaii 4" were sterilized and planted in MS medium in a growth chamber with a 16 h light (22 °C) and 8 h dark (20 °C) cycle. After about 2 months of growth, the seedlings were transplanted to a substrate of perlite, vermiculite and peat soil and grown under a natural photoperiod in a glass house. Clean roots, stems, leaves, petioles and stolons were taken from 24 strawberry plants in the woodland during growth. Due to limited organs, we collected stolon tips, fully open flowers and immature fruits from about 100 strawberry plants. Samples were rapidly frozen in liquid nitrogen after collection and stored at −80 °C for subsequent RNA extraction. Notably, due to the limited number of samples, only two biological replicates were available for the stolon tips and fully open flowers, while all other samples had three biological replicates [39].

After sequencing the transcriptome, the raw data after transcriptome sequencing were obtained clean after removing certain sequences: those with adaptor sequences, more than 10% of bases that were not clear and low quality sequences. Indexing of the reference genomes was performed with HISAT2 v2.1.0 (http://daehwankimlab.github.io/hisat2/ accessed on 16 May 2019). The reference genomes were downloaded from the GDR website. The clean data were aligned with the diploid strawberry reference genome using HISAT2 v2.1.0 with default parameters. Finally, the reads for each gene were calculated by HTSeq v0.6.1 (https://github.com/simon-anders/htseq accessed on 16 May 2019), as well as the fragment per kilobase of the exon model per million mapped fragments (FPKM) value for each gene [39].

The FPKM values in different tissues (Flower, leaf, Petiole, Stolon, Stolon tip, Root, Stem, Immature fruit) and different stresses (Cold and Heat) were selected from our own transcriptome data [39], the reads per kilobase per million (RPKM) values used in anther, carpel, cortex, pith, embryo, ghost, ovule, seed, wall, Rg (15D and 22D) and Yw (15D and 22D) were downloaded from Li et al. [40], and the FPKM values of *FveBBX7* and *FveBBX16* in "Ruegen" strawberry fruits were obtained from Gu et al. [41]. All the transcriptomic data were transformed at the log2 level. Finally, heat maps were created using TBtools1.116 software.

### 2.6. Plant Materials and Photoperiodic Treatments

For photoperiodic experiments, seedlings of two-month-old woodland strawberry were moved into artificially illuminated incubators and subjected to two photoperiods, LD (day 16 h/night 8 h) and SD (day 8 h/night 16 h), respectively. Strawberry leaves were collected and the expression profiles of *FveBBX* genes under different photoperiodic conditions were examined. Strawberry leaves were collected every 4 h during each 24 h period, with three biological replicates in each collection [42].

For the cold- and heat-treatment experiments, strawberry seeds were similarly sterilized and planted in MS medium and grown in a growth chamber. After 2 months of growth, the seedlings were transferred to two other growth chambers which were kept at 40 °C for heat treatment and at 4 °C for cold treatment. Whole strawberry seedings from treatments at 0 h, 2 h, 6 h, 12 h, 24 h and 48 h were collected and stored in liquid nitrogen in a −80 °C refrigerator after fast freezing for RNA extraction and three biological replicate experiments.

### 2.7. RNA Extraction, Quantitative Real-Time PCR Analysis

Samples stored in a −80 °C refrigerator were extracted from total RNA using modified CTAB [43]. cDNA was then synthesized from 1000 ng of total RNA using the Hiscript II

Q-RT SuperMix for qPCR kit (Vazyme, Nanjing, China) according to the manufacturer's instructions. The 4×gDNA Wiper Mix and 5×HiScript II qRT SuperMix II were used for the removal of genomic DNA and reverse transcription, respectively. A 20 μL system was prepared with twenty-times-diluted cDNA, 0.4 μL primer 1, 0.4 μL primer 2, 10 μL 2×ChamQ SYBR qPCR Master Mix (Vazyme) and dd water. The results of gene qRT-RCR were obtained by using a CFX-96 real-time fluorescence quantitative PCR instrument (Bio rad, Hercules, CA, USA) with one reaction at 95 °C for 30 s, 40 cycles at 95 °C for 10 s and 60 °C for 30 s, another reaction at 65 °C for 5 s and a final reaction at 85 °C for 5 min. Three biological replicates and two technical replicates were performed for each sample. We used the *FveActin* gene as an internal reference gene and primer sequences for all genes are available in the Table S1. Relative levels of gene expression were determined using the $2^{-\Delta\Delta CT}$ method [44].

## 3. Results

### 3.1. Identification and Characterization of BBX Proteins in the Woodland Strawberry

We queried and searched the latest genome database of the woodland strawberry to identify BBX proteins. A total of 22 BBX proteins were identified and named according to their positions on the chromosome. These BBX proteins had different lengths of coding sequences with different isoelectric points and molecular weights (Table 1). In particular, *FveBBX20* had the shortest coding sequence, followed by *FveBBX6* and *FveBBX17*, while *FveBBX11* had the longest coding sequence, and *FveBBX13* and *FveBBX15* were the second and third, respectively. The molecular weights of these genes were proportional to the lengths of the coding sequence. Most of the strawberry BBX proteins were acidic, and their isoelectric points were less than seven. However, three members of the BBX protein family (FveBBX16, FveBBX17 and FveBBX20) had an isoelectric point greater than seven, indicating that they were basic. Among them, FveBBX8 had the smallest isoelectric point (4.30), while FveBBX16 had the largest isoelectric point (8.99) (Table 1).

**Table 1.** *BBX* genes in the woodland strawberry.

| Protein Name | Gene ID | CDS | Protein/AA | pI | MW | Introns | Chr Srart | Chr End | Chrom |
|---|---|---|---|---|---|---|---|---|---|
| FveBBX1 | FvH4_1g12110.t3 | 675 | 225 | 6.24 | 24,772.05 | 3 | 6615641 | 6618432 | Fvb1 |
| FveBBX2 | FvH4_2g10070.t1 | 639 | 213 | 6.17 | 23,958.62 | 1 | 8897464 | 8899003 | Fvb2 |
| FveBBX3 | FvH4_2g24910.t1 | 1182 | 394 | 5.74 | 43,576.53 | 3 | 20252750 | 20254655 | Fvb2 |
| FveBBX4 | FvH4_2g38990.t1 | 942 | 314 | 6.62 | 34,463.57 | 2 | 28109150 | 28111588 | Fvb2 |
| FveBBX5 | FvH4_2g41420.t2 | 939 | 313 | 6.33 | 34,506.73 | 1 | 29305073 | 29306617 | Fvb2 |
| FveBBX6 | FvH4_3g03640.t1 | 552 | 184 | 4.37 | 20,264.61 | 1 | 2045154 | 2047400 | Fvb3 |
| FveBBX7 | FvH4_3g17750.t1 | 864 | 288 | 5.12 | 30,812.60 | 2 | 11299422 | 11301298 | Fvb3 |
| FveBBX8 | FvH4_3g21230.t1 | 744 | 248 | 4.30 | 27,112.50 | 1 | 14224305 | 14225858 | Fvb3 |
| FveBBX9 | FvH4_4g08980.t1 | 1071 | 357 | 5.63 | 38,993.97 | 1 | 10078076 | 10079592 | Fvb4 |
| FveBBX10 | FvH4_4g10930.t1 | 717 | 239 | 4.71 | 26,293.62 | 3 | 14596852 | 14598977 | Fvb4 |
| FveBBX11 | FvH4_4g23090.t1 | 1437 | 479 | 5.66 | 51,772.28 | 3 | 25667711 | 25670086 | Fvb4 |
| FveBBX12 | FvH4_4g26540.t1 | 912 | 304 | 6.25 | 33,281.15 | 2 | 27896068 | 27898487 | Fvb4 |
| FveBBX13 | FvH4_4g26550.t1 | 1416 | 472 | 5.62 | 52,959.81 | 4 | 27899275 | 27901406 | Fvb4 |
| FveBBX14 | FvH4_4g27390.t1 | 1353 | 451 | 5.56 | 50,333.90 | 1 | 28409309 | 28411538 | Fvb4 |
| FveBBX15 | FvH4_5g12150.t1 | 1374 | 458 | 5.56 | 51,231.43 | 1 | 6852249 | 6854527 | Fvb5 |
| FveBBX16 | FvH4_6g37140.t1 | 795 | 265 | 8.99 | 28,638.63 | 0 | 29197326 | 29198120 | Fvb6 |
| FveBBX17 | FvH4_6g37790.t1 | 585 | 195 | 8.30 | 20,963.21 | 2 | 29750190 | 29757327 | Fvb6 |
| FveBBX18 | FvH4_6g40380.t1 | 1254 | 418 | 5.45 | 45,246.46 | 5 | 31931413 | 31935765 | Fvb6 |
| FveBBX19 | FvH4_6g43570.t1 | 1335 | 445 | 5.43 | 48,826.65 | 4 | 33682838 | 33686104 | Fvb6 |
| FveBBX20 | FvH4_6g43580.t1 | 339 | 113 | 8.75 | 12,618.81 | 0 | 33686668 | 33687006 | Fvb6 |
| FveBBX21 | FvH4_6g44270.t1 | 759 | 253 | 4.71 | 27,901.25 | 2 | 34125442 | 34128499 | Fvb6 |
| FveBBX22 | FvH4_6g45860.t1 | 1158 | 386 | 5.35 | 42,345.92 | 1 | 35108088 | 35110508 | Fvb6 |

CDS, coding sequence; AA, amino acid residues; pI, isoelectric point; MW, molecular weight; Chr, chromosome.

### 3.2. Conserved Domain and Motif Analysis of BBX Proteins in the Woodland Strawberry

In order to predict the functional conservation and differentiation of FveBBX proteins, we analyzed the conserved domain and motif composition of 22 BBX proteins in the woodland strawberry. FveBBX proteins with similar structures were clustered in each group (Figure 1A). Among these, 10 FveBBX proteins had one B-box domain at the N-terminus, 4 of which contained a CCT domain at the C-terminus. The other 12 proteins all contained two B-box domains, and 7 contained CCT domains at the C-terminus (Figure 1C). The specific positions of the three conserved structural domains of FveBBX proteins are shown in Figure S1.

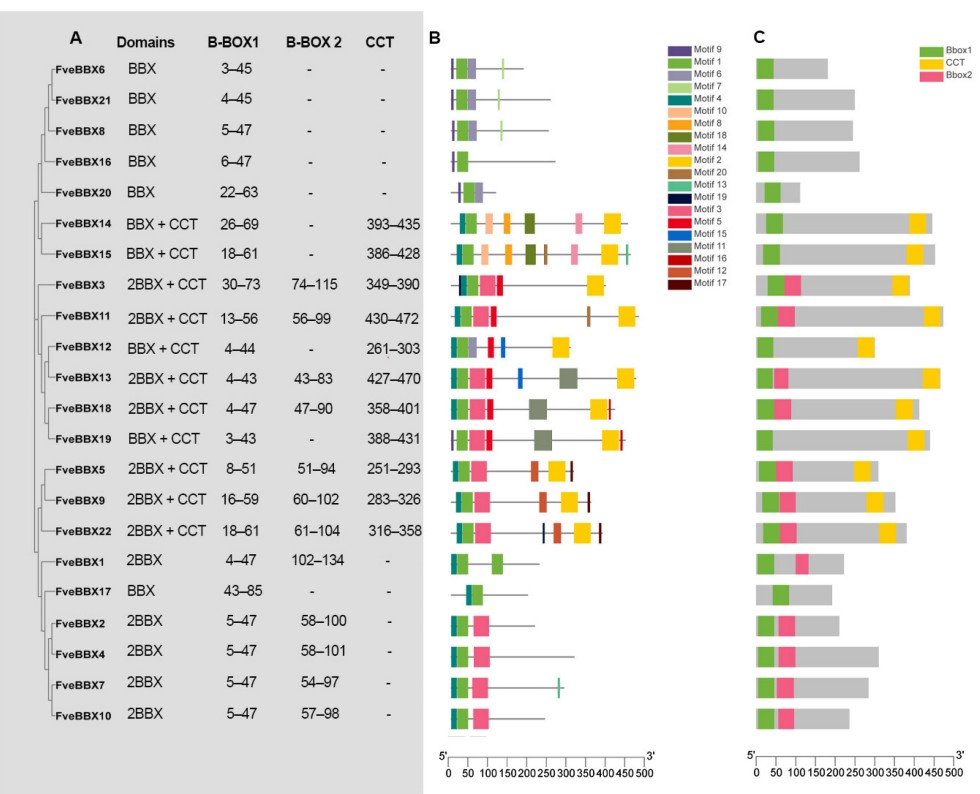

**Figure 1.** Domain and motif analysis of BBX proteins in the woodland strawberry. (**A**) Phylogenetic tree of the BBX proteins in the woodland strawberry with the location and length of each domain. (**B**) Motif composition of BBX proteins in the woodland strawberry. Different motifs were shown in different colors. The distribution of conservative patterns is shown in Supplementary Figure S2. (**C**) Conserved domain composition of BBX proteins in the woodland strawberry.

In addition, we analyzed 20 motifs of FveBBX proteins (Figure 1B). The logos of Motifs 1–20 are shown in Figure S2. The results showed that, depending on the width, Motif 11 was the largest, followed by Motif 2 and Motif 3. Motif 1 was found in all FveBBX proteins. Notably, Motif 2 and Motif 3 were present in 11 and 12 FveBBX proteins, respectively. Motif 1 and Motif 3 had similar conserved amino acids, and Motif 2 was also highly conserved in the FveBBX proteins.

### 3.3. Phylogeny and Gene Structure of the FveBBX Gene Family

To further elucidate the evolutionary relationships and functional differences of the 22 *FveBBX* genes, we used MEGA7.0 software to compare the full-length sequences of these FveBBX proteins and constructed a phylogenetic tree (Figure 2A). The results of the phylogenetic analysis were consistent with the former findings in rice, *Arabidopsis* and other crops, where the *BBX* family was divided into five subfamilies. In subfamily I and subfamily II, the vast majority of FveBBX proteins contained two B-box structural

domains and one CCT structural domain, except for FveBBX19 and FveBBX12, who had only one B-box structural domain. Subfamily III contained two B-box structural domains, with the exception of FveBBX17, which had only one B-box structural domain. All members of subfamily IV contained a B-box structural domain and a CCT structural domain, while members of subfamily V contained only a B-box structural domain. To better understand the classification of the woodland strawberry BBX members, phylogenetic trees of the woodland strawberry FveBBX proteins and *Arabidopsis* AtBBX proteins were constructed (Figure S3), and all BBX proteins were similarly divided into five subfamilies, with BBX members from *Arabidopsis* and the woodland strawberry clustered together. In addition, we also constructed phylogenetic trees for the B1 (Figure 2B), B2 (Figure 2C) and CCT (Figure 2D) structural domains of the woodland strawberry to further understand the phylogenetic relationships. The FveBBX proteins containing the B1 structural domain were divided into five subfamilies, and the FveBBX proteins containing the B2 or CCT structural domains were divided into three subfamilies.

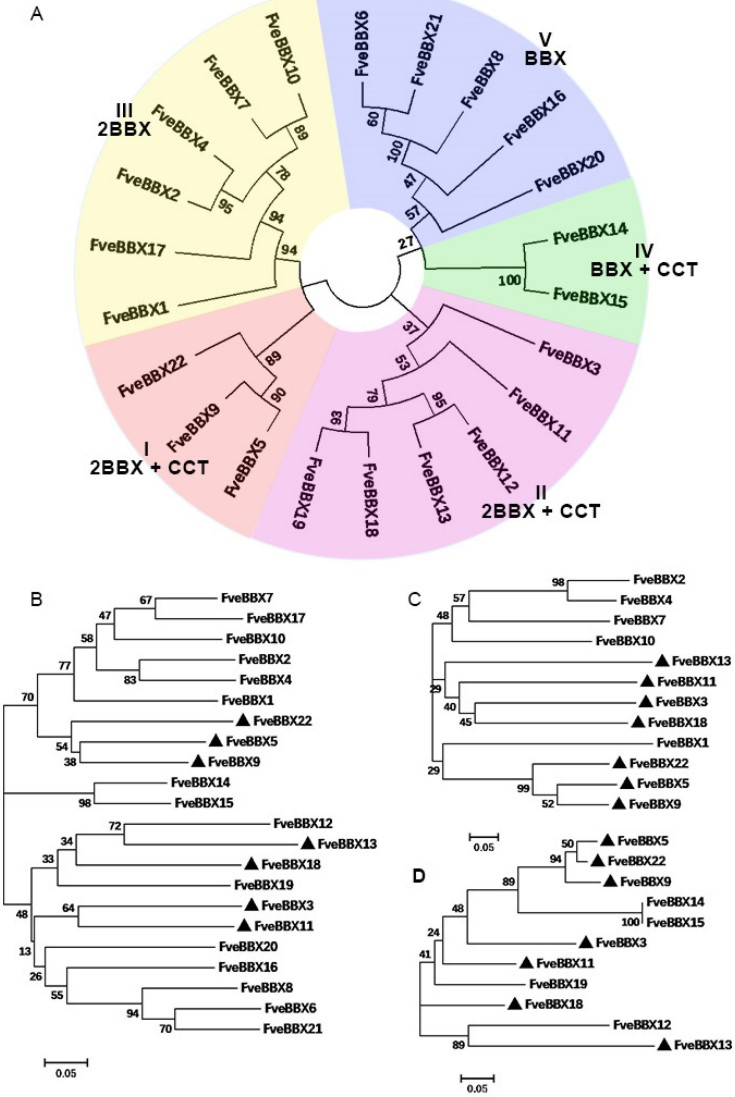

**Figure 2.** Phylogenetic analysis of *BBX* genes in the woodland strawberry. The trees shown are based on the alignments of the full-length protein sequences (**A**), B-box1 (**B**), B-box2 (**C**) and CCT (**D**) domains. MEGA7.0 software was used to perform 1000 bootstrap replicates, and the phylogenetic tree was constructed by a neighbor-joining method. The scale bar represents 0.05 amino acid substitutions per site. Members marked with black triangles have two B-box domains and a CCT domain.

The genetic structure of *BBX* members was mapped using the GSDS website (Figure S4). Except for *FveBBX16* and *FveBBX20*, which had no introns, the other *BBX* members contained anywhere from one to five introns. Notably, *BBX* genes in the same subfamily had a similar gene structure. For example, *FveBBX14* and *FveBBX15* in subfamily IV both had only one intron, and *FveBBX5*, *FveBBX9* and *FveBBX22* in subfamily I all contained two exons. Therefore, the structure of the genes may have changed during evolution, providing insight into the evolutionary relationships of the *FveBBX* genes.

*3.4. Cis-Elements in the Promoters of FveBBX Genes*

We extracted the promoter region of the *FveBBX* genes (2 kb of genomic DNA sequence before the translation start site) and used the PlantCare database to predict their cis-acting elements, finally identifying 29 cis-acting elements (Tables 2, S2 and S3). In addition to the conventional elements, the CAAT-box and TATA-box, which were present in all *FveBBX* genes, the other cis-acting elements could be classified into four types: hormone corresponding elements, abiotic stress corresponding elements, light-response elements and response elements involved in plant growth and development. Among them, the CAT-box is involved in the expression in the meristem, the O2-site is a regulatory element of maize lysozyme metabolism, and the GCN4_motif is involved in the regulation of endosperm expression (Table S3).

**Table 2.** Cis-acting element identification in *FveBBX* genes.

| Gene Name | Hormone Responsive | | | | | Light Responsive | Stress Responsive | | | | Other Responsive |
|---|---|---|---|---|---|---|---|---|---|---|---|
| | ABA | GA | IAA | MeJA | SA | | Anaerobic | Drought | Low-Temperature | Defense | |
| *FveBBX1* | + | + | + | + | | + | + | + | | | + |
| *FveBBX2* | | + | | + | | + | + | | | | + |
| *FveBBX3* | + | + | | + | + | + | + | | + | + | + |
| *FveBBX4* | + | | + | + | | + | + | | | + | |
| *FveBBX5* | | + | + | + | + | + | + | | | | |
| *FveBBX6* | + | + | + | + | + | + | + | + | | | |
| *FveBBX7* | + | | + | + | | + | + | + | | | + |
| *FveBBX8* | + | | + | + | | + | + | + | + | + | + |
| *FveBBX9* | + | + | + | + | | + | + | | | | + |
| *FveBBX10* | + | | + | + | + | + | + | | | | |
| *FveBBX11* | | + | + | + | | + | + | + | | | + |
| *FveBBX12* | + | + | | + | | + | + | | | | + |
| *FveBBX13* | + | + | + | + | | + | + | + | | + | + |
| *FveBBX14* | + | + | + | + | | + | + | + | + | | + |
| *FveBBX15* | | + | | + | | + | + | | + | + | |
| *FveBBX16* | + | + | | + | | + | + | | + | | + |
| *FveBBX17* | + | + | + | + | | + | + | + | | | + |
| *FveBBX18* | | + | + | + | | + | + | + | + | | + |
| *FveBBX19* | + | | | + | + | + | + | + | + | | + |
| *FveBBX20* | + | + | + | | + | + | + | | | | + |
| *FveBBX21* | + | + | | + | + | + | + | + | | | + |
| *FveBBX22* | + | + | + | | | + | | + | | + | + |

The symbol "+" indicates the presence of this cis-acting element

As shown in the Table 2, all 22 *FveBBX* gene promoters contained light-response elements (G-box, Box 4, GT1-motif, AE-box, GATA-motif, TCT-motif, MRE, I-box, TCCC-motif, Sp1, ATCT-motif). Among the hormone-response elements, 17 genes contained abscisic-acid-response elements (ABRE), 17 genes contained gibberellin-response elements (P-box, GARE-motif, TATC-box), 15 genes contained growth-hormone-response elements (TGA-element, AuxRR-core), 20 genes contained MeJA-response elements (CGTCA-motif, TGACG-motif), and 7 genes contained salicylic-acid-response elements (TCA-element). In addition, there were 4 abiotic stress corresponding elements, ARE, MBS, LTR and TC-rich repeats, present in the promoters of the *FveBBX* genes 21, 12, 7 and 6, respectively. These results suggested that the *BBX* family of the woodland strawberry has the potential to function in abiotic stresses, hormone-response pathways and the light response.

### 3.5. Expression Analysis of FveBBX Genes in Various Tissues and Organs

We collected flowers, leaves, petioles, stolons, stolon tips, roots, stems and immature fruits of the "Hawaii 4" woodland strawberry for transcriptomic analysis (Figure 3A; Table S5). FPKM was used to calculate the relative expression levels of *BBX* genes, and a value greater than one indicated expression. The results showed that 20 *FveBBX* genes were expressed in at least one tissue, and *FveBBX17* and *FveBBX19* were expressed in all organs at levels of less than one. These *FveBBX* genes showed different expression patterns in different organs, with *FveBBX10* and *FveBBX11* being expressed in all organs at higher levels, and *FveBBX2*, *FveBBX12* and *FveBBX16* showing low levels in all organs. A few *FveBBX* genes showed tissue specificity; for example, *FveBBX3*, *FveBBX14* and *FveBBX15* were most highly expressed in leaves, and *FveBBX6* was preferentially expressed in stems. *FveBBX3* and *FveBBX21*, which were mostly expressed in flowers and barely in the other seven organs, could be important for the woodland strawberry bloom. The highest *FveBBX6* and *FveBBX11* expression levels were in stems, whereas *FveBBX22* was more frequently expressed in petioles. There was a strong expression of *FveBBX3*, *FveBBX9*, *FveBBX10*, *FveBBX14*, *FveBBX15* and *FveBBX18* in leaves but medium expression in petioles or stolons. These results suggested that *FveBBX* genes might play different roles in different tissues.

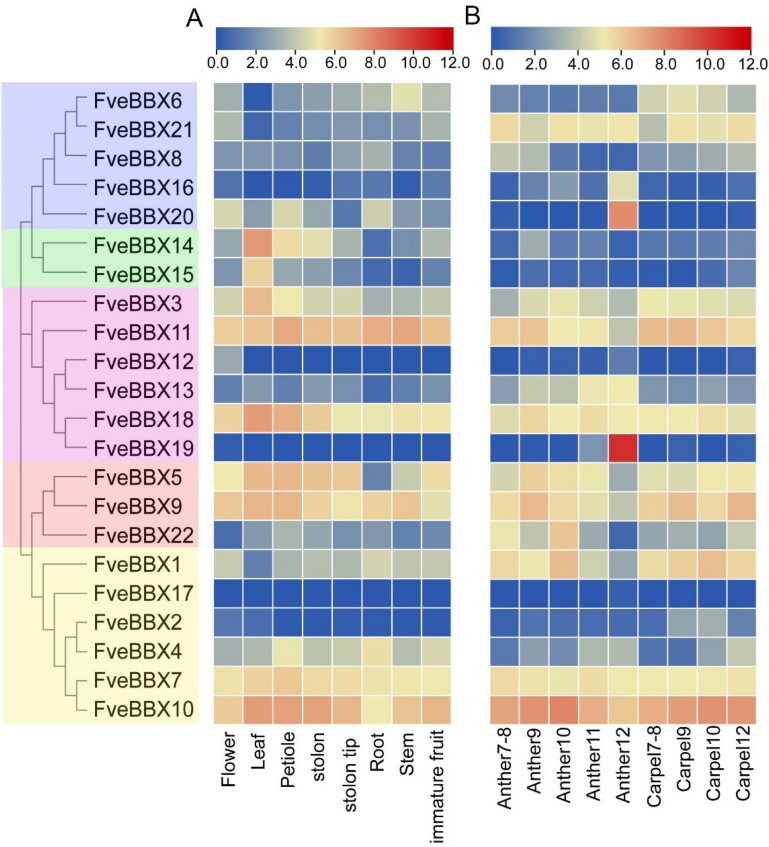

**Figure 3.** Expression profile of *FveBBX* genes in different tissues and organs of the woodland strawberry. (**A**) Expression of *FveBBX* genes in the open flower, leaf, petiole, stolon, stolon tip, root, stem and immature fruit of the woodland strawberry. (**B**) The expression of 22 *FveBBX* genes in anthers and carpels. The heat map is shown at log2 and clustered according to the phylogenetic relationship level. Redder colors indicate higher expression. The numbers after the anthers and carpels represent the different periods of the third stage of flower development, details of which are shown in Supplementary Table S4.

There are 13 developmental stages in the strawberry flower, which are broken up into three sections: Stages 1–4, early flower development; Stages 5–7, initiation and development of reproductive organs; and Stages 8–13, differentiation of floral organs. From

previously published data, we downloaded the expression of the woodland strawberry *BBX* genes in anthers and carpels during the third stage of flower development (Figure 3B; Table S6) [40]. The *FveBBX* genes had different expression patterns, with 18 genes expressed in at least one of these stages. During anther development, the expression of *FveBBX4* and *FveBBX13* increased at least seven-fold in Anther 12 compared to Anther 7. The expression of *FveBBX16*, *FveBBX19* and *FveBBX20* also increased markedly in Anther 12, but barely at all in other stages, especially *FveBBX19*. In total, 6 genes (*FveBBX1*, *FveBBX3*, *FveBBX5*, *FveBBX9*, *FveBBX10*, *FveBBX11*) showed at least a two-fold decrease in expression in Anther 12. Furthermore, 12 genes had higher levels of expression in Anther 12 than in Anther 7 during carpel development, with *FveBBX4*, *FveBBX8*, *FveBBX21* and *FveBBX22* showing at least a two-fold increase, especially *FveBBX4* with an increase of more than fourteen-fold. The expression levels of the *FveBBX* genes did, however, fluctuate significantly during the development of the anther and carpel. For instance, the expression levels of *FveBBX6*, *FveBBX11* and *FveBBX18* in the carpels and *FveBBX1*, *FveBBX3*, *FveBBX5*, *FveBBX10*, *FveBBX11*, *FveBBX18* and *FveBBX22* in the anthers revealed a tendency of "rising and then dropping". It is evident that the *BBX* genes in the woodland strawberry play several roles in carpel and anther development.

### 3.6. Expression Analysis of BBX Genes during the Development and Ripening of Woodland Strawberry Fruit

The pre-fruit development period of woodland strawberries can be divided into five stages, they represent pre-fertilization (Stage 1), 2–4 days post-anthesis (Stage 2), 6–9 days post-anthesis (Stage 3), 8–10 days post-anthesis (Stage 4) and 10–13 days post-anthesis (Stage 5). The wall and seeds together form the achenes, and the seeds are subdivided into embryos and ghosts (seeds without embryos). The seeds in Stage 1 are ovules (Figure 4A; Table S7) [40]. According to published transcriptome data [40], three genes (*FveBBX6*, *FveBBX13* and *FveBBX16*) showed at least a 1.8-fold increase in expression, while four genes (*FveBBX7*, *FveBBX8*, *FveBBX9* and *FveBBX21*) showed at least a 1.5-fold decrease from Stage 1 to Stage 2 in seeds, indicating that fertilization affects the expression of *FveBBX* genes. Five genes (*FveBBX1*, *FveBBX5*, *FveBBX9*, *FveBBX10* and *FveBBX16*) showed a steady drop in expression in embryos at Stages 3–5, whereas the expression of the other five genes (*FveBBX4*, *FveBBX7*, *FveBBX11*, *FveBBX13* and *FveBBX18*) showed a gradual rise. The expression of *FveBBX3* increased at Stage 4 and decreased at Stage·5. The expression pattern of *BBX* genes in ghosts was different from that in embryos; in addition to *FveBBX11*, *FveBBX13* and *FveBBX21*, the expression of *FveBBX3* and *FveBBX20* increased gradually. Five genes (*FveBBX1*, *FveBBX9*, *FveBBX10*, *FveBBX16* and *FveBBX22*) had increasing and then decreasing expression patterns, and three genes (*FveBBX4*, *FveBBX6* and *FveBBX7*) showed the reverse pattern. Interestingly, *FveBBX6*, *FveBBX20* and *FveBBX22* were specifically expressed in ghosts but not in embryos. The expression patterns of *FveBBX* genes in the walls at Stages 1–5 were markedly different. There were five genes (*FveBBX3*, *FveBBX4*, *FveBBX5*, *FveBBX10* and *FveBBX18*) whose expression increased, one gene (*FveBBX8*) whose expression declined, eight genes (*FveBBX1*, *FveBBX6*, *FveBBX13*, *FveBBX14*, *FveBBX15*, *FveBBX16*, *FveBBX20* and *FveBBX22*) whose expression increased and then decreased, and three genes (*FveBBX7*, *FveBBX11* and *FveBBX21*) whose expression decreased and then increased. Notably, *FveBBX14* and *FveBBX15* were barely expressed in seeds, suggesting that they may be specifically involved in wall development.

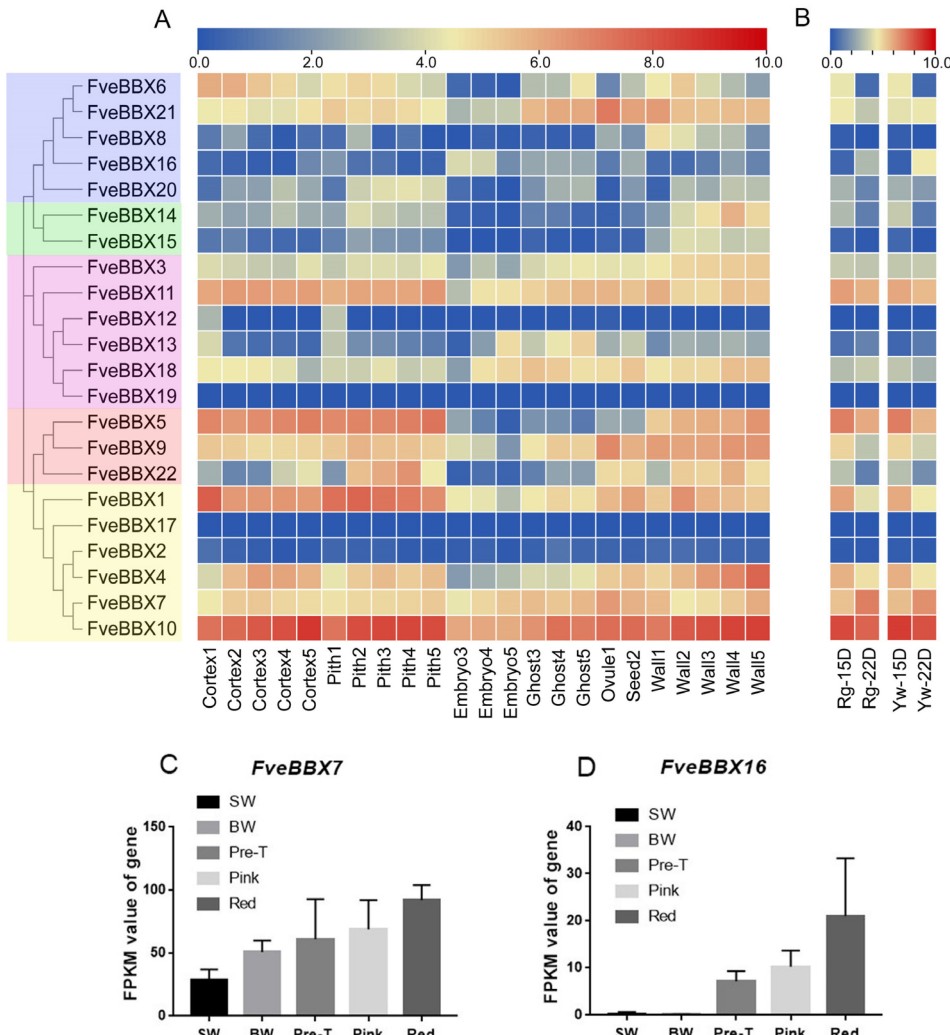

**Figure 4.** Expression profile of *FveBBX* genes in developing and ripening of strawberry fruit. (**A**) *FveBBX* genes' expression patterns in developing fruit tissues, which including embryo, ovule, ghost, wall, cortex and pith. (**B**) Changes in the expression of 22 *FveBBX* genes in immature and mature fruits of "Ruegen" and "Yellow Wonder". (**C**,**D**) Changes in the expression of *FveBBX7* and *FveBBX16* in "Ruegen" strawberry fruits. Numbers after tissues represent different developmental stages: Stage 1 (pre-fertilization), Stage 2 (2–4 days post-anthesis (DPA)), Stage 3 (6–9 DPA), Stage 4 (8–10 DPA) and Stage 5 (10–13 DPA). Rg, Ruegen; Yw, Yellow Wonder; 15D, 15 DPA; 22D, 22 DPA. SW, small white stage; BW, big white stage; pre-T, pre-turning stage; pink, pink stage; red, red stage.

The receptacle of the woodland strawberry consists of a cortex and a pith [40]. In total, 18 genes in the cortex and 19 genes in the pith were expressed at levels greater than one in at least one of the five stages. Certain *FveBBX* genes had similar expression patterns in the cortex and pitch. For example, *FveBBX12* was specifically expressed only at Stage 1 of cortex and pith development, whereas *FveBBX20* was specifically not expressed at Stage 1. The expression of a small number of genes changed significantly during the five stages. *FveBBX10* expression increased steadily during cortex development, whereas *FveBBX6* expression continuously declined. During pitch development, the expression of *FveBBX5* and *FveBBX7* increased steadily, whereas that of *FveBBX14* decreased gradually. In addition, the expressions of most genes fluctuated up and down, showing one or two peaks during cortex and pitch development. Notably, the expression changes in *FveBBX9* and *FveBBX22* were the opposite in the cortex and pith. Therefore, the various *FveBBX* genes play positive or negative roles in the early development of woodland strawberry fruit.

Most studies on ripening aspects of strawberries, especially late ripening, have been related to anthocyanins. However, anthocyanins did not accumulate in "Hawaii 4"; thus, we examined the expression profile of the *BBX* gene in "Hawaii 4" fruit only in immature fruit. In order to further investigate the role of *BBX* genes in the later stages of fruit ripening, we extracted the expression data of *BBX* genes from the transcriptome data of two other woodland strawberry cultivars, "Ruegen" (red fruits) and "Yellow Wonder" (pale yellow fruits), at 15 and 22 days of fruit development [40]. As shown in Figure 4 and Table S8, 15 *BBX* genes were expressed in at least one stage in "Ruegen" fruits, among which *FveBBX6* was only expressed in immature fruits, while *FveBBX13* and *FveBBX16* were at low levels only in mature fruits. The expression levels of 12 genes (*FveBBX1*, *FveBBX3*, *FveBBX4*, *FveBBX5*, *FveBBX6*, *FveBBX9*, *FveBBX10*, *FveBBX11*, *FveBBX14*, *FveBBX20*, *FveBBX21* and *FveBBX22*) decreased and those of 4 genes (*FveBBX7*, *FveBBX13*, *FveBBX16* and *FveBBX18*) increased after fruit ripening. In "Yellow Wonder" fruit, 16 *BBX* genes were expressed in at least one stage, among which *FveBBX6* and *FveBBX14* were preferentially expressed in immature fruit, whereas *FveBBX13* and *FveBBX16* were expressed only in mature fruit like in "Ruegen" strawberry fruit. While the expression levels of most *FveBBX* genes decreased, those of *FveBBX3*, *FveBBX7* and *FveBBX21* increased. It is worth noting that during fruit ripening, the expression levels of *FveBBX3* and *FveBBX18* increased in "Ruegen" and slightly decreased in "Yellow Wonder", and that of *FveBBX21* showed the opposite pattern, indicating that *FveBBX* genes play different roles in the ripening of strawberry fruits. After fruit ripening, the expressions of *FveBBX7*, *FveBBX13* and *FveBBX16* increased, particularly *FveBBX7* and *FveBBX16*, in both varieties of strawberry. As a result, we used previously published "Ruegen" strawberry transcriptome data to validate these findings [41]. According to the data, the expression level of *FveBBX7* (Figure 4C; Table S9) climbed steadily as the fruit ripened, while the expression level of *FveBBX16* (Figure 4D; Table S9) was initially low but quickly rose as the fruit ripened. The results of the two experiments were consistent, suggesting that *FveBBX7* and *FveBBX16* may play an important role in strawberry ripening.

*3.7. Expression Analysis of FveBBX Genes under Cold and Heat Stresses*

Numerous studies have demonstrated the involvement of BBX proteins in signaling pathways induced by abiotic stresses. "Hawaii 4" seedlings were exposed to 4 °C of cold and 40 °C of heat, respectively, to test the role of the *FveBBX* genes in strawberry resistance to abiotic stress. As shown in Figure 5, certain genes showed similar expression profiles under different stresses, while others showed different expression patterns, which might indicate that *FveBBX* genes have functional differences in strawberry resistance to cold and heat stress. For example, six *FveBBX* genes (*FveBBX2*, *FveBBX6*, *FveBBX12*, *FveBBX13*, *FveBBX17* and *FveBBX19*) were generally expressed at low levels, whereas *FveBBX10* and *FveBBX20* were expressed at high levels.

The expression of five *FveBBX* genes (*FveBBX3*, *FveBBX7*, *FveBBX10*, *FveBBX16* and *FveBBX18*) were significantly elevated under heat treatment (Figure 5A; Table S10). Among them, the expression of *FveBBX7* and *FveBBX16* increased more than 2-fold, that of *FveBBX7* increased 5.32-fold at 48 h and that of *FveBBX16* increased 8.3-fold at 24 h. The expression of five genes (*FveBBX1*, *FveBBX6*, *FveBBX13*, *FveBBX15* and *FveBBX20*) decreased, that of *FveBBX1* by more than 4-fold. The majority of genes, however, had zigzag variations in expression at different treatment intervals, and not all genes displayed steady increases or decreases in expression. For example, the expression of *FveBBX5*, *FveBBX8* and *FveBBX14* increased more than 2-fold at 2 h but gradually decreased after 12 h.

Under cold treatment, the expression levels of 11 genes increased (*FveBBX1*, *FveBBX2*, *FveBBX4*, *FveBBX7*, *FveBBX8*, *FveBBX15*, *FveBBX16*, *FveBBX18*, *FveBBX20*, *FveBBX21* and *FveBBX22*), and those of *FveBBX20*, *FveBBX21* and *FveBBX22* increased at least 2-fold, with a peak at 6 h (Figure 5B). The expressions of four genes (*FveBBX5*, *FveBBX6*, *FveBBX11* and *FveBBX14*) decreased, with that of *FveBBX14* decreasing at least 11-fold at 12 h. Similar to heat treatment, a few genes showed tortuous changes in expression.

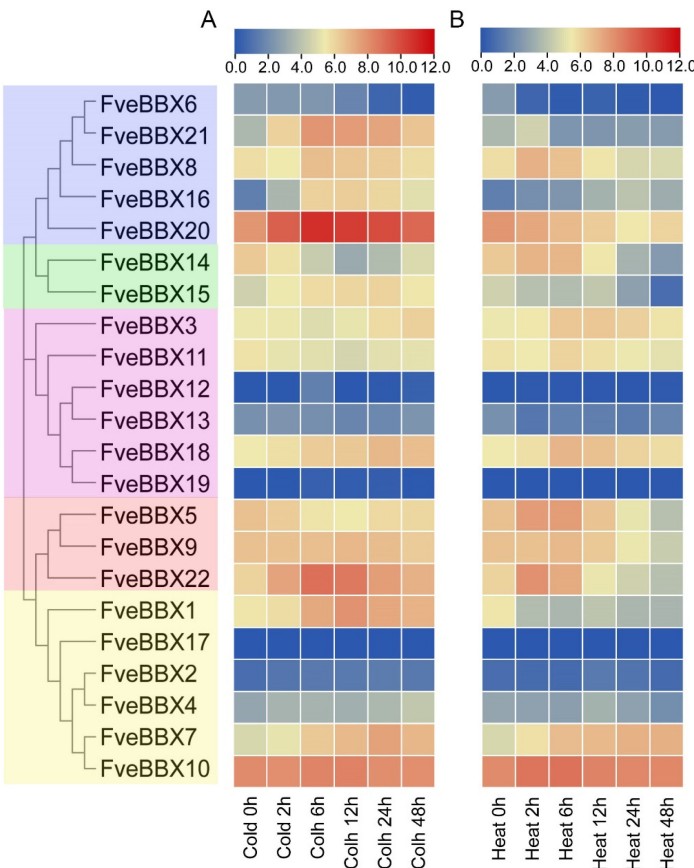

**Figure 5.** Expression of *FveBBX* genes of the woodland strawberry under cold and heat stresses. (**A**) Expression of *FveBBX* genes of the woodland strawberry under heat stress. (**B**) Expression of *FveBBX* genes of the woodland strawberry under cold stress.

Additionally, we discovered that the increase or decrease in the expression level of the same gene differed under different treatments; for instance, five genes (*FveBBX1*, *FveBBX15*, *FveBBX20*, *FveBBX21* and *FveBBX22*) showed decreased expression following heat treatment but increased expression following cold treatment. These findings implied that these genes have diverse functions in strawberry resistance to heat and cold stressors.

*3.8. Analysis of BBX Gene Expression in the Woodland Strawberry during Different Photoperiods*

RNA-seq analysis provides a global view of *FveBBX* genes expression levels. From the heat map, it was found that most of the *FveBBX* genes were preferentially expressed in leaves and petioles, and leaves were the most responsive organ of the plant to light. We quantitatively analyzed the *BBX* genes using strawberry leaves as materials to explore the impact of the photoperiod on *FveBBX* genes. *FveBBX3* (Figure 6A), *FveBBX7* (Figure 6B) and *FveBBX10* (Figure 6C), which were highly expressed in leaves, were sampled and analyzed at 4 h intervals under LD (16 h light/8 h dark) and SD (8 h light/16 h dark) conditions. As shown in Figure 6, the three *BBX* genes showed similar expression patterns under LD conditions: a gradual decrease during the light period, an increase at the onset of the dark period, a peak at the end of the dark period, a marked decrease at the end of the dark period and a nadir at the beginning of the light period. Under SD conditions, the transcripts of the three *BBX* genes were much more abundant during the dark period than during the light period. The expression levels of *FveBBX3* and *FveBBX10* decreased during the light period, whereas that of *FveBBX7* increased. At the beginning of the dark period, the expression levels of the three genes increased significantly, but they decreased or stagnated at 20 h. The expression of *FveBBX3* peaked at 16 h, whereas those of *FveBBX7* and *FveBBX10* peaked at the end of the dark period and then decreased, as under the

LD condition. Notably, *FveBBX3* and *FveBBX10* showed similar expression trends during the first 16 h under different photoperiodic treatments, whereas *FveBBX7* genes showed opposite expression patterns during the first 16 h.

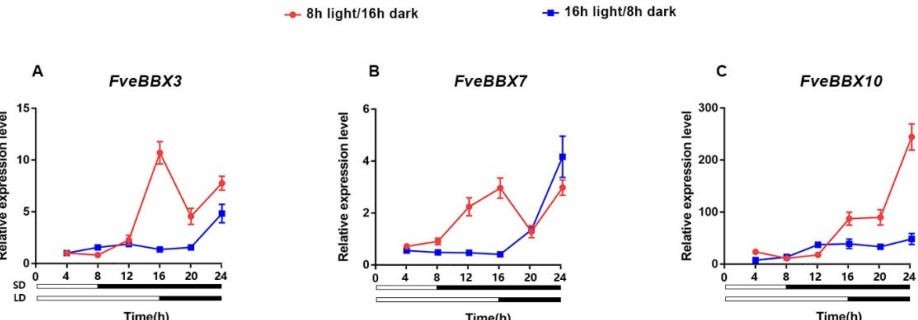

**Figure 6.** Daily expression pattern of *FveBBX* gene under LD and SD conditions analyzed by qRT-PCR. Expression patterns of *FveBBX3* (**A**), *FveBBX7* (**B**) and *FveBBX10* (**C**) under different photoperiod. The white and black bars at the bottom indicate light and dark periods, respectively. The standard deviation of the independent biological duplicate experiments is shown by the error bar.

## 4. Discussion

### 4.1. Evolution of the BBX Gene Family in the Woodland Strawberry

Compared to previous studies [45], we identified an additional gene with two B-box domains and one CCT domain and named it *FveBBX22*. This is likely a result of updates to the woodland strawberry genome database. In all, 22 *FveBBX* genes were identified, markedly fewer than the 32, 64, 37 and 31 *BBX* genes in *Arabidopsis*, apple, pear, and tomato, respectively. This may be related to fragmentary and tandem replication of genes [46]. For example, we found 22 homologous pairs of genes in apple, only one of which was involved in fragmentary replication in the strawberry *CO* family (*FveCO2/7*) [42]. In addition, different species have different *BBX* gene combinations. In the woodland strawberry, 9 genes had only one B-box domain, and 13 genes had two. In grape [8], apple [9], pear [10], sweet cherry [11], potato [12] and tomato [13], 17, 30, 23, 10, 18 and 19 genes contained two B-box domains, and 7, 34, 13, 5, 12 and 12 genes contained one B-box gene, respectively. Therefore, B-box sequences are largely conserved in plants, and the BBX genes of several species may have descended from a common ancestor.

*FveBBX* genes with two or one B-box domain and one CCT domain were classified as subfamily I, II or IV. *FveBBX12* and *FveBBX19* in subfamilies I and II had only one B-box domain. Although *FveBBX17* belonged to subfamily III, it contained only one B-box domain. The remaining genes with only one B-box domain belonged to subfamily V. We have found similar phenomena in other species. For example, *PavBBX15* in the sweet cherry has only two B-box structural domains but belongs to subfamily I [11]. In the pear, *PbBBX4*, which has only one B-box domain and no CCT domain, belongs to subfamily I [10]. BBX proteins initially had only one B-box structural domain, B-box2. B-box1 was replicated from B-box2, and then a CCT structural domain was added to the carboxyl end of the protein, resulting in BBX proteins with two B-box structural domains and one CCT structural domain. At the same time, there was a deletion event of the B-box2 structural domain during evolution, which eventually resulted in BBX proteins with only one B-box structural domain or one B-box and one CCT structural domain [5]. Therefore, we hypothesize that *FveBBX12*, *FveBBX19* and *FveBBX17* may have undergone evolutionary deletion or replication during evolution.

### 4.2. Potential Role of FveBBX Gene in Strawberry Responses to Abiotic Stresses and Environmental Stimulus

We identified 29 cis-acting elements in the promoter of *FveBBX* genes. Among them, the frequency of photoresponsive elements was highest, but we also identified hormone corresponding elements and abiotic-stress-response elements. These results are consistent

with those of *Arabidopsis* and rice. We also found that *FveBBX* genes' expression could be modulated by the photoperiod. This is consistent with reports that *BBX14* (*FveCO2*), *BBX22* (*FveCO3*), *BBX5* (*FveCO4*), *BBX18* (*FveCO5*), *BBX15* (*FveCO7*) and *BBX9* (*FveCO8*) showed different diurnal expression patterns under different photoperiods. The expression profiles of *BBX22* (*FveCO3*) and *BBX18* (*FveCO5*) were the opposite under LD and SD conditions [42]. *FveBBX3*, *FveBBX7* and *FveBBX10*, which were highly expressed in leaves, showed significant expression changes at 24 h under LD and SD conditions. *FveBBX* genes exhibit different expression patterns under different photoperiods. Therefore, *FveBBX* genes might promote strawberry adaptation to changes in the photoperiod, but the underlying mechanisms are unknown.

In addition to the influence of light in the natural environment, temperature is also a major factor affecting the growth of plants. High temperatures in summer or low temperatures in winter can affect the nutritional and reproductive growth of the strawberry. *BBX* genes are involved in responses to abiotic stresses, some by responding to light signals [47]. In *Arabidopsis*, *AtBBX1* is not only involved in the regulation of flowering as the first verified *BBX* gene but also promotes drought escape by accelerating flowering and setting seeds under drought conditions [48]. The salt resistance of *Arabidopsis* is influenced by *AtBBX24* [33]. The overexpression of *AtBBX18* in *Arabidopsis* reduces germination and survival under heat stress and suppresses the expression of heat-stress-related genes. This suggests that *AtBBX18* negatively regulates heat tolerance in *Arabidopsis* [32]. In addition, *OsBBX8* overexpression in rice causes leaf tip etiolation and upregulates several genes associated with aging, suggesting that *OsBBX8* decreases drought resistance [49]. To investigate the role of *BBX* genes in the resistance of the woodland strawberry to abiotic stress, we subjected 22 *FveBBX* genes to high- and low-temperature treatments. Our results showed that at least 16 *FveBBX* genes were regulated by abiotic stress and that different stress treatments induced or repressed *FveBBX* genes to different degrees. Under cold stress, most genes whose promoters had low-temperature-response elements were upregulated. In this work, it was discovered that cold and heat treatment increased the expressions of *FveBBX7*, *FveBBX16* and *FveBBX18*, suggesting that these three genes may have advantageous functions under various conditions. It is possible that *FveBBX1*, *FveBBX15*, *FveBBX20*, *FveBBX21* and *FveBBX22* have separate or opposing mechanisms underlying their responses to abiotic stresses, because heat and cold decreased and increased their expression, respectively. Among them, the expression of *FveBBX1*, a homolog of *Arabidopsis AtBBX18*, decreased gradually under heat stress, indicating that *FveBBX1* reduces, by an as-yet-unknown mechanism, the tolerance of the woodland strawberry to high temperature. The study of the response properties of *FveBBX* genes under different abiotic stresses will help to enhance our understanding of the functions of *BBX* genes in different abiotic stress signaling pathways.

### 4.3. The Potential Role of BBX Gene in Strawberry Growth and Development

*BBX* genes contribute to various stages of plant growth and development, including light morphogenesis, flowering induction, response to shade [50] and anthocyanin synthesis [51]. We analyzed the expressions of 22 *FveBBX* genes in eight tissues and two stages (nutritional and reproductive growth), as well as during fruit development. The results showed that *BBX* genes play a variety of roles during strawberry vegetative and reproductive growth. Among them, certain *FveBBX* genes may have unique or redundant functions in particular tissues or at particular developmental stages.

*FveBBX2* and *FveBBX21*, which were only expressed in flowers, may be involved in flower formation in the woodland strawberry. Indeed, *VvBBX10* and *VvBBX11* are highly expressed in grape floral tissues, implicating these genes in flower development [8]. *BBX* genes are involved in the growth of pollen tubes. *PbBBX5* expression is significantly increased in pear pollen tubes [52], suggesting a function in their growth. In this study, the expressions of *FveBBX16*, *FveBBX19* and *FveBBX20* were high at Stage 12 of anther development, suggesting involvement in anther growth and/or senescence. *AtBBX21* is

expressed in dry and germinating seeds and it is able to repress the transcriptional activity of *ABI5* and regulate ABA, thereby mediating the germination of *Arabidopsis* seeds [53]. In the process of seed formation, the *FveBBX* gene expression of the woodland strawberry had similar or obvious differences at different developmental stages of the embryo, ghost and wall. For example, during fertilization, the *FveBBX6* expression in Seed 2 increased nearly five-fold; and the expression quantity of *FveBBX1*, *FveBBX7*, *FveBBX13*, *FveBBX14*, *FveBBX15*, *FveBBX20* and *FveBBX22* in the wall from Wall 1 to Wall 2 suggested that changes of at least 20% had taken place. These results suggested that multiple *FveBBX* genes may be involved in seed development in the woodland strawberry.

*4.4. The Potential Roles of Strawberry BBX Genes in the Development and Ripening of Fruit*

Plants undergo a variety of changes when fruit ripens, most notably in terms of color [54]. The fruit color of strawberries is an important characteristic of strawberries due to the accumulation of anthocyanins and their derivatives [55]. Additionally, anthocyanins are crucial for secondary wall development, UV resistance and pathogen defense [56]. An increasing number of studies have shown that *BBX* genes are involved in anthocyanin accumulation and fruit development and maturation. For example, AtBBX22 interacts with HY5 to promote the expression of anthocyanin pigment 1 (PAP1) and with anthocyanin-biosynthesis genes to promote anthocyanin synthesis [57]. The overexpression of MdBBX33, a homolog of *Arabidopsis* AtBBX22, results in elevated anthocyanin levels in *Arabidopsis* seedlings [58]. In addition, MdBBX37 interacts with MdMYB1, MdMYB9 and MdHY5 to coregulate anthocyanin biosynthesis [59]. The same study was performed in pears, demonstrating that the dimer formed by PpBBX18 and PpHY5 interacts with PpBBX21 to suppress anthocyanin biosynthesis [60]. MaCOL1 expression significantly increases during natural and ethylene-induced ripening of bananas [61]. Interactions between SlBBX19, SlBBX20 and SlBBX26 and a ripening inhibitor (SlRIN) regulate fruit ripening in tomato [62]. As a result, *BBX* genes may be crucial for fruit development.

According to this study, the expressions of *FveBBX7* and *FveBBX16* were low or absent in immature fruits but high in mature fruits, implicating these genes in fruit ripening. This is similar to previous research; the *FaBBX22* gene in the octaploid strawberry, which is homologous to *FveBBX7*, depends on the FaBBX22–FaHY5 heterodimer to promote anthocyanin accumulation in white-meat strawberry and thus participated in fruit ripening [63]. Similar to *FveBBX7*, *FveBBX16* might have a similar purpose in fruit; however, because it was expressed at low levels early in fruit development, *FveBBX16* might be crucial for fruit ripening. However, the expression levels of *FveBBX1*, *FveBBX4*, *FveBBX5*, *FveBBX6*, *FveBBX9* and *FveBBX10* were low in mature fruits and high in immature fruits, suggesting that these six genes had a negative regulatory function in the ripening of woodland strawberry fruits. Regardless, further research is still required to determine the specific process.

**5. Conclusions**

In this study, 22 *FveBBX* genes were identified and systematically analyzed in the woodland strawberry, including conserved domains and motifs, phylogenetic relationships, gene structure, cis-acting elements, and expression pattern analysis. Multiple types of cis-acting elements were found in the *FveBBX* promoter sequence, indicating that *FveBBX* genes are involved in plant responses to light, hormones and stress. The transcriptome data of *FveBBX* genes from different organs and developmental stages and under different conditions indicated that *FveBBX* genes may have a variety of roles in the growth and development of the woodland strawberry. Transcriptome and qRT-PCR indicated that *FveBBX* genes may play an important role in fruit development and maturation. In conclusion, the genome-wide analysis of *FveBBX* genes will provide a foundation for the functional analysis of *BBX* genes in strawberry, and our data may help to select suitable candidate genes in future studies.

**Supplementary Materials:** The following supporting information can be downloaded at: https://www.mdpi.com/article/10.3390/horticulturae9070842/s1, Figure S1. Multiple sequence alignment of the BBX proteins from the woodland strawberry. The multiple sequence alignment results of B-box1 (A), B-box2 (B) and CCT (C) domains in 22 FveBBX proteins are shown. The sequences were compared using Jalview2.10.4b1. Figure S2. Motif composition of the BBX family in the woodland strawberry. The MEME website was used to analyze the motifs of BBX proteins, and the number of motifs was set as 20. The height of each letter indicates the conservation of each residue in all identified proteins. Figure S3. Phylogenetic analysis of BBX proteins in strawberry and *Arabidopsis thaliana*. ClustalW in MEGA7.0 was used to compare the full-length BBX protein sequences of the woodland strawberry (FveBBX) and *Arabidopsis thaliana* (AtBBX), and phylogenetic trees were constructed using the adjacency method of MEGA7.0. The red triangle represents the woodland strawberry BBX protein, and the blue circle represents the *Arabidopsis* BBX protein. Figure S4. Genetic structure analysis of 22 *FveBBX* genes in the woodland strawberry. The genetic structure of FveBBX genes was mapped using the GSDS website. Blue boxes represent the 5′ or 3′ UTR, red boxes represent the CDS, and black lines represent introns. Table S1. Primers used for qPCR analysis. Table S2. Number of cis-acting elements in the *BBX* gene promoter of the woodland strawberry. Table S3. Cis-elements identified in *FveBBX* genes. Table S4. The details of the development stages. Table S5. Expression profile of *FveBBX* genes in different organs and tissues of the woodland strawberry. Flowers, leaves, petioles, stolons, stolon tips, roots, stems and immature fruits were collected for transcriptome analysis. The gene expression results were expressed by FPKM value. Table S6. Expression profile of *FveBBX* genes in anthers and carpels of the woodland strawberry Table S7. Expression profile of *FveBBX* genes in developing fruit tissues of the woodland strawberry. Table S8. Expression profile of *FveBBX* genes in early-stage fruits and ripening fruits of the woodland strawberry. Table S9. Changes in the expression of *FveBBX7* and *FveBBX16* in "Ruegen" strawberry fruits. Table S10. The FPKM value of the *BBX* genes in woodland strawberry seedlings treated with cold and heat.

**Author Contributions:** Conceptualization, H.W. (Han Wu) and H.C.; writing—original draft preparation, D.X.; methodology, D.X., H.W. (Hongkun Wang), X.F., Y.M., Y.H. and Y.W.; software, D.X., H.W. (Hongkun Wang) and X.F.; writing—review and editing, H.W. (Han Wu), H.C., D.X. and J.D.; project administration, H.W. (Han Wu) and H.C. All authors have read and agreed to the published version of the manuscript.

**Funding:** This work was supported by the National Natural Science Foundation of China (31972371).

**Data Availability Statement:** The *Arabidopsis* and woodland strawberry *BBX* gene family databases were downloaded from the TAIR (https://www.arabidopsis.org/ accessed on 16 April 2022) website and GDR (https://www.rosaceae.org/species/fragaria/fragaria_vesca accessed on 16 April 2022) website, respectively. In this study, RNA-seq data on the carpel, anther, and fruit developmental tissues of woodland strawberry were obtained from the published data set (https://doi.org/10.1038/s41438-019-0142-6 accessed on 26 May 2022). The other RNA-seq data presented in this study are available in the Supplementary Materials.

**Conflicts of Interest:** The authors declare no conflict of interest.

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
