# Peer review of "Genome-Wide Identification, Phylogenetic and Expression Analysis of the B-Box Gene Family in the Woodland Strawberry (Fragaria vesca)"

_horticulturae, doi:10.3390/horticulturae9070842_

Round 1
Reviewer 1 Report
The manuscript is well-written and is on the scope of the journal. The authors showed a preliminar characterization and analysis of expression profile of B-BOX 2 transcription factor family in woodland strawberry, a subject with high interest to readers in the same area.
Minor comments
Line 127. Was phylogenetic tree analysis carried using the whole protein sequence or using a specific domain. Whole sequence introduces an additional variable, protein length, which will create a separation of proteins regarding their length. Is correct called it as a phylogenetic analysis? Please clarify it.
Line 142. Please add information about sampling to obtain sequencing data, as well as, number of replicates and sequencing procedure. The reference provided by the authors doesn’t present it.
Line 152. Add description of growth of plants under stresses (cold and heat).
Reviewer 2 Report
The manuscript entitled “EGenome-wide identification and expression analysis of B-BOX 2 transcription factor family in woodland strawberry ” seems very interesting, well written, and has novelty. After careful evaluation, I believe that this manuscript should be considered for Publication. I have a few Major suggestions for the improvement of the manuscript.
Title
Please make the title of the article more understandable. Also add the Latin name of the plant.
Summary
Add the Latin name.
Put some numerical data from the results of the study in the summary.
Entrance
A few paragraphs should be added to the summary section about strawberry in terms of herbal properties, production value and human health.
Too many abbreviations are used in the abstract, write their full names where they are first used.
In the introduction, studies on other plants are shown as examples and their Latin names are written. No need for Latin.
material and method
Explain in detail the content of RNA analysis kits, pcr cycles and pcr components
Line 174. The sentence should be started with a capital letter.
The manufacturers and countries of the package programs used in the study should be written.
The explanations under the figures in the study are too long, shorten them.
A total of 77 references were used in the study. shorten it a bit.
I will review it again after necessary revisions.
Reviewer 3 Report
This paper deals with the investigation of the transcription factor genes classified into B-Box family in woodland strawberry. The contents of this paper include the gene expression profiles based on the published papers. I am confused because the different papers used the different cultivars. In the present paper, they were mixed and are not mentioned. Do the different cultivars have different physiological features? Do the different physiological features affect gene expression profiles?
Reviewer 4 Report
The review of the article entitled “Genome-wide identification and expression analysis of B-BOX 2 transcription factor family in woodland strawberry”.
In this study, researchers analyzed the BBX gene family in woodland strawberries and its potential role in various biological processes. They identified 22 B-BOX2 genes in woodland strawberries and analyzed their phylogeny and structure, as well as their expression in different developmental stages and tissues. Authors investigate also changes in the expression of genes encoding these transcription factors in response to cold stress and heat stress as well as in different photoperiods. The work is very extensive and contains a large number of interesting results which are discussed in the context of present knowledge, but several aspects need improvement.
1. Most of the analyses were performed on the “Hawai 4” cultivar, however during investigation the expression levels of genes during different stages of development results referring to two other cultivars were added. The purpose of this is not clear to me, especially since it remains not clear if their expression patterns differ from those observed in the “Hawai 4” cultivar.
2. The methodology of experiments related to developmental stages and tissues is difficult to track since authors refer to additional materials for the details. This information should be placed in the main text of the manuscript and the figures should be self-explanatory. This applies especially to the criterion for assessing the degree of fruit ripeness. Also, even a schematic illustration, showing which changes in the expression level were analyzed would be also very helpful. While the exact number can be included in the supplementary materials, such basic information as abbreviation and methodology of establishing developmental stages should be available in the main text.
3. The conditions of experiments involving abiotic stresses are not clear. The authors mention that the plants were exposed to 4°C and 40°C but the time of exposure, the developmental stage of plants during exposition as well as the tissue investigated were not mentioned.
In my opinion, after introducing these corrections, the work can be published.
Round 2
Reviewer 2 Report
Ms is ready for publication
Reviewer 3 Report
Thank you for the response. I am convinced.